# Accuracy of preferred language data in a multi-hospital electronic health record in Toronto, Canada

Camron D. Ford[1,2], Thomas Bodley[1,3,4,5], Martin Betts[1,3,4,5], Rob A. Fowler[4,5,6,7], Alexis Gordon[1,4], Michele James[1], Shail Rawal[4,8], Christina Reppas-Rindlisbacher[4,6,8,9], Paul Tam[1], George Tomlinson[6,8], Christopher J. Yarnell[1,3,4,5]*

1 Scarborough Health Network, Toronto, Canada, 2 Temerty Faculty of Medicine, University of Toronto, Toronto, Canada, 3 Scarborough Health Network Research Institute, Toronto, Canada, 4 Department of Medicine, University of Toronto, Toronto, Canada, 5 Interdepartmental Division of Critical Care Medicine, University of Toronto, Toronto, Canada, 6 Institute for Health Policy, Management, and Evaluation, University of Toronto, Toronto, Canada, 7 Department of Critical Care Medicine, Sunnybrook Health Sciences Centre, Toronto, Canada, 8 University Health Network, Toronto, Canada, 9 Sinai Health System, Toronto, Canada

* cyarnell@shn.ca

## Abstract

Accurate preferred language data is a prerequisite for providing high-quality care. We investigated the accuracy of preferred language data in the electronic health record (EHR) of a large community hospital network in Toronto, Canada. We conducted a point-prevalence audit of patients admitted to intensive care, internal medicine, and nephrology services at three hospitals. We asked each patient "What is your pre-ferred language for health care communication?" and reported on agreement (with 95% confidence intervals [CI]) between interview-based and EHR-based preferred language. We used Bayesian multilevel logistic regression to analyze the association between patient factors and the accuracy of the EHR for patients who preferred a non-English language. Between June 17, 2024, and July 19, 2024, we interviewed 323 patients, of whom 124 (38%) preferred a non-English language. Median age was 77 years and 46% were female. EHR accuracy was 86% for all patients. The proba-bility of the EHR correctly identifying a patient with non-English preferred language (sensitivity) was 69% (CI 60–77), specificity was 97% (CI 94–99), positive predictive value was 95% (CI 88–98), and negative predictive value was 83% (CI 79–87). There were 26 different non-English preferred languages, most commonly Cantonese (27%) and Tamil (14%). Accuracy was better for patients who were female or older, and var-ied by hospital and medical service. Mechanisms to improve accuracy for language preference data are needed to improve the validity of research studying preferred language, mitigate algorithmic bias, and overcome language-based inequities.

**Data availability statement:** Deidentified data for the primary analysis and code are posted publicly here: https://github.com/cjyarnell/PreferredLanguageAccuracy.

**Funding:** This work was supported by the Canadian Critical Care Trials Group Network of Networks Summer Student Award (CF), Sunnybrook Program to Access Research Knowledge (CF), the Nephrology Program at the Scarborough Health Network (CF). JP Bickell Foundation Medical Research Grant (CJY). Mak Pak Chiu and Mak-Soo Lai Hing Chair in General Internal Medicine, University of Toronto (SR). The funders had no role in study design, data collection and analysis, decision to publish, or preparation of the manuscript.

**Competing interests:** The authors have declared that no competing interests exist.

## Author summary

Clinical outcomes such as mortality rate or length of hospitalization are worse for patients who cannot communicate in the language of their healthcare team, especially if interpretation is not used. Many healthcare systems and research projects rely on routinely collected preferred language data to plan their interpretation strategies and study the association between preferred language and the processes or outcomes of care. However, the accuracy of this data is uncertain, especially in settings where there are multiple possible preferred languages. We interviewed 323 inpatients across three hospitals in Toronto, Canada, asking them "What is your preferred language for health care communication?" and comparing their response to the preferred language charted in the electronic health record (EHR). We found that 124 (38%) preferred a non-English language. There were 26 different non-English preferred languages. EHR accuracy was 86% for all patients, and the probability of the EHR correctly identifying a patient with non-English preferred language (sensitivity) was 69% (CI 60–77). Mechanisms to improve accuracy for preferred language preference data are needed to ensure we provide high-quality care to all patients, regardless of their language ability.

## Introduction

Patients who do not speak the language in which their care is provided are at higher risk of harm and poor outcomes. Canadian data shows an 18–30% higher risk of harm for hospital inpatients who do not speak English, compared to inpatients who do speak English [1,2]. Such patients are common in Canada, where 2021 census data showed 690,000 people who did not speak either official language. In the United States, 5,720,834 (4.4%) households were classified as "limited English-speaking" in the 2023 American Community Survey [3]. Neither statistic includes people who may speak basic English, but would prefer to discuss complicated medical decisions in another language. Language-appropriate care, where a patient receives care in their preferred language for healthcare communication via shared fluency or professional interpretation, can help address potential disparities [4].

Language-appropriate care is essential for effective care. It helps clinicians to understand patient symptoms and values in order to make informed recommendations, and it helps patients to understand their situation and options, in order to make informed decisions [1,2]. Ensuring access to communication, independent of a patient's preferred language, is a pillar of equitable care [5,6]. Non-English language preference could also become a source of algorithmic bias, whereby artificial intelligence models perpetuate the bias encoded in training data by incorporating it into their predictions [7]. Given the importance of language-appropriate care, clinicians and healthcare systems need accurate data regarding their patients' preferred language.

The accuracy of preferred language data in electronic health records (EHRs) varies. A study conducted in 2020 in Toronto, Canada, found sensitivities of 81% and 12% at two academic hospital sites [8]. Another study conducted in an outpatient setting in the USA found that the EHR accurately identified preferred language in two-thirds of Spanish-speaking patients [9]. Previous Canadian evaluations involved only academic hospitals and uncommon EHRs [10]. Identifying preferred language is especially relevant in Toronto, Canada, where 42.5% of the population have a mother tongue other than English or French [11]. To inform quality improvement and research efforts aimed at ensuring equitable care across language proficiencies, we evaluated the accuracy of the preferred language field in a commonly used EHR (Epic Systems) at three large community hospitals.

## Methods

We performed a prospective point-prevalence audit of inpatients at three community hospitals within the same health network in Toronto, Canada, to evaluate the accuracy of the preferred language field in the EHR. We used an approach similar to prior work [8]. At the health network we studied, asking about preferred language is routine and mandated during patient registration for appointments or hospital admissions. The patient's preferred language is then entered into the EHR. The audit was deemed quality improvement by the Scarborough Health Network research ethics board, which granted a waiver of the requirement for informed consent (PQI-24-002).

The only inclusion criterion was admission to the internal medicine ward, nephrology ward, or intensive care unit. The only exclusion criterion was inability to determine whether their preferred language was English. One investigator (CF) prospectively interviewed consecutive patients admitted to the intensive care unit, internal medicine ward, and nephrology ward across the three hospitals. These services were chosen due to the role language-appropriate care plays in each service. Patients found on all three services can have complex, multi-system conditions that require frequent communication between patient and clinical team, and important decisions regarding goals of care and other interventions. Together, these services make up a large proportion of inpatients across the three hospital sites.

On each interview day, they selected a particular geographic ward of the hospital containing approximately 30–50 ward rooms and proceeded sequentially interviewing the patients in each room. This continued each day until the entire ward service had been sampled. After introducing themselves and the project to the patient, they asked one question: "What is your preferred language for health care communication?" [8] We recorded the response as "English" or "Not English." In the case where the preferred language was not English, we recorded the preferred language.

A patient was considered unavailable if they were not present at the time the interviewer passed by their room. A response was considered unable to be elicited if the patient was incapacitated, intubated, or unable to understand the question on language preference. If the patient was unavailable at the time of interview, or a response could not be elicited, a substitute decision maker (SDM) was contacted and their response to the question was recorded. If an SDM was unable to be reached upon the first attempt, a second attempt was made. After an unsuccessful second attempt, the patient was excluded from the audit.

We recorded interview-based preferred language and EHR-based preferred language in a REDCap database [12]. We also collected age, sex, hospital of admission, most responsible service, and length of stay. We compared the absolute difference in percentage of patients preferring a non-English language between interview-based and EHR-based methods using a two-sided test of proportions.

In the primary analysis, we calculated percentage agreement and Cohen's kappa; we also reported the sensitivity, specificity, positive predictive value, negative predictive value of using EHR-based non-English preferred language to identify true non-English language preference. We used a McNemar's test to assess whether discordant responses were more likely to prefer a non-English language. As a sensitivity analysis, we repeated the primary analysis in subgroups defined by whether we obtained interview-based preference from the patient or their SDM.

The secondary analysis focused on identifying factors associated with correctly identifying non-English preferred language. We did this by modeling the probability of an EHR-based non-English preference among patients who had an interview-based non-English preference using multilevel Bayesian logistic regression clustering by preferred language [13].

Fixed effect predictors were age, sex, hospital, length-of-stay, and inpatient service; preferred language was a random effect, to allow for the incorporation of languages with few respondents without causing overfitting [14]. Hospitals were modeled as fixed effects because there were only 3, with known differences in the distribution of languages. Prior distributions for fixed effects were skeptical (normal(0, 0.5) in log-odds space). We modeled age in decades to make its odds ratio more interpretable. We modeled the logarithm (base 2) of length-of-stay to aid in model convergence. We reported ORs, including the median OR which measures the extent of variability between clusters (languages) [15,16].

We used a Bayesian approach because when clusters are small, frequentist multilevel models may have difficulties with convergence [14]. In addition, a multilevel approach clustering by preferred language allows low-frequency languages to be included in the regression, without ignoring broad trends across non-English languages or collapsing them into an "Other" category [17]. There were no sensitivity analyses incorporating alternative prior distributions. We reported uncertainty using 95% confidence intervals (CI) in the primary analysis, and 95% credible intervals (CrI) in the secondary analysis. Models were fit in R using the brms package, using 4 chains, each with 1000 warmup and 2000 model fit iterations [18,19].

## Results

Between June 17, 2024, and July 19, 2024, we identified 349 patients across 3 hospitals, of whom 26 were excluded because we could not contact them or their SDM. No contacted patients or SDMs declined participation. That left 323 patients who were interviewed (Table 1). The number of included patients per hospital ranged from 85 to 108. The median age was 77 years (Interquartile range [IQR]: 65–85) and 46% of patients were female. The most common responsible service was internal medicine. For 132 patients (41%), we obtained preferred language information from the substitute decision-maker.

The number of patients who preferred a non-English language was 93 (29%) by EHR, and 124 (38%) by interview (Table 2, absolute difference of 9.6%, CI 2.4 to 16.8, $p < 0.01$). There were 26 different languages represented among the interview-based non-English preferred languages. The most common non-English preferred languages were Cantonese (26.6%), Tamil (13.7%), and Italian (8.1%).

The interview-based and EHR-based preferred language agreed in 278 (86%) of patients and Cohen's kappa was 0.69 (CI 0.61 to 0.77) (Table 3). The probability of correctly identifying a patient with non-English preferred language

**Table 1. Characteristics of Patients.**

| Patient Characteristic | Hospital Site | | | Total |
|---|---|---|---|---|
| | 1<br>n = 130 | 2<br>n = 85 | 3<br>n = 108 | N = 323 |
| Female (%) | 56 (43%) | 41 (58%) | 53 (49%) | 150 (46%) |
| ICU (%) | 31 (24%) | 22 (26%) | 6 (6%) | 59 (18%) |
| Nephrology (%) | 26 (20%) | 0 | 0 | 26 (8%) |
| Medicine (%) | 73 (56%) | 63 (74%) | 102 (94%) | 238 (74%) |
| Median age, years (IQR) | 73 (63 − 83) | 75 (63-82) | 82 (71-90) | 77 (65-85) |
| Median length of hospital stay, days (IQR) | 10 (4 − 25) | 7 (4 −19) | 8 (4-24) | 9 (4 − 23) |

**Caption:** List of audit sample characteristics grouped by hospital site. ICU = intensive care unit. IQR = interquartile range.

**Table 2. Concordance between interview- and EHR-based preferred language responses.**

| | | Interview-Based Preferred Language | | |
| --- | --- | --- | --- | --- |
| | | **English** | **Non-English** | |
| EHR documented language | English | 192 | 38 | 230 |
| | Non-English | 7 | 86 | 93 |
| | | 199 | 124 | 323 |

**Caption:** Dichotomized table (English and Non-English) of inpatient preferred spoken language data from interviews (columns) and EHR (rows). Sensitivity: 69% (CI 60–77). Specificity: 97% (CI 94–99). Positive predictive value: 95% (CI 88–98). Negative predictive value: 83% (CI 79–87). CI = 95% confidence interval. The positive condition is a non-English language preference.

**Table 3. Non-English preferred languages.**

| Languages Surveyed | Frequency Among Non-English Languages | Number of Respondents |
| --- | --- | --- |
| Cantonese | 27% | 33 |
| Tamil | 14% | 17 |
| Italian | 8% | 10 |
| Tagalog | 4% | 5 |
| Bengali | 4% | 5 |
| 21 other languages | 44% | 23 |

**Caption** The five most frequently recorded non-English languages (all ≥ 4%) during the audit. *Other languages, counts suppressed because they had fewer than 5 respondents: Albanian, Amharic, Arabic, Armenian, Dari, Farsi, Greek, Gujarati, Hakka, Hindi, Japanese, Macedonian, Mandarin, Marathi, Punjabi, Spanish, Tigrinya, Toisanese, Turkish, Urdu, Vietnamese.*

(sensitivity) was 69% (CI 60–77); specificity was 97% (CI 94–99), positive predictive value was 95% (CI 88–98), and negative predictive value was 83% (CI 79–87). The true preference in patients with discordance was more likely to be non-English language (OR 5.43 [CI 2.43 to 12.17], $p < 0.001$).

Interview-based non-English language preference was less common when the patient was the source compared to when the SDM was the source (50/191 [26%] vs 58/132 [44%], absolute difference 17.8% [CI 4.2% to 25.2%, $p < 0.01$]). Whether interview-based preferred language was obtained from patients or SDMs, accuracy was similar (169/181 [88%] vs 109/132 [83%], $p = 0.13$) and Cohen's kappa was similar (0.68 vs 0.67).

The Bayesian analysis revealed several factors potentially associated with the EHR correctly identifying a non-English language preference (Fig 1). Model diagnostics were reassuring, with all R-hat values equal to 1, no divergences, and effective sample sizes all greater than 2500. Correct identification of preferred language was less likely if a patient was male compared to female (OR 0.54, CrI 0.28 to 1.02, OR < 1 with probability 97%), cared for in Hospital 2 (OR 0.58, CrI 0.28 to 1.21, OR < 1 with probability 92%) or Hospital 3 (OR 0.68, CrI 0.35 to 1.33, OR < 1 with probability 87%) compared to Hospital 1, or cared for by the internal medicine service (OR 0.63, CrI 0.31 to 1.30, OR < 1 with probability 90%) compared to nephrology and critical care. By contrast, correct identification of preferred language was more likely for older patients (OR per decade 1.29, CrI 0.94 to 1.81, OR > 1 with probability 94%).

There was minimal variation in odds of correct identification across non-English preferred languages. The median odds ratio, which quantifies the median change in odds of correct identification when switching between two non-English preferred languages, was 1.31 (CrI 1.01 to 1.99). However, the credible intervals for the odds ratio associated with each individual language were wide (Fig 2), showing significant residual uncertainty.

## Odds ratios for correctly identifying non-English language preference

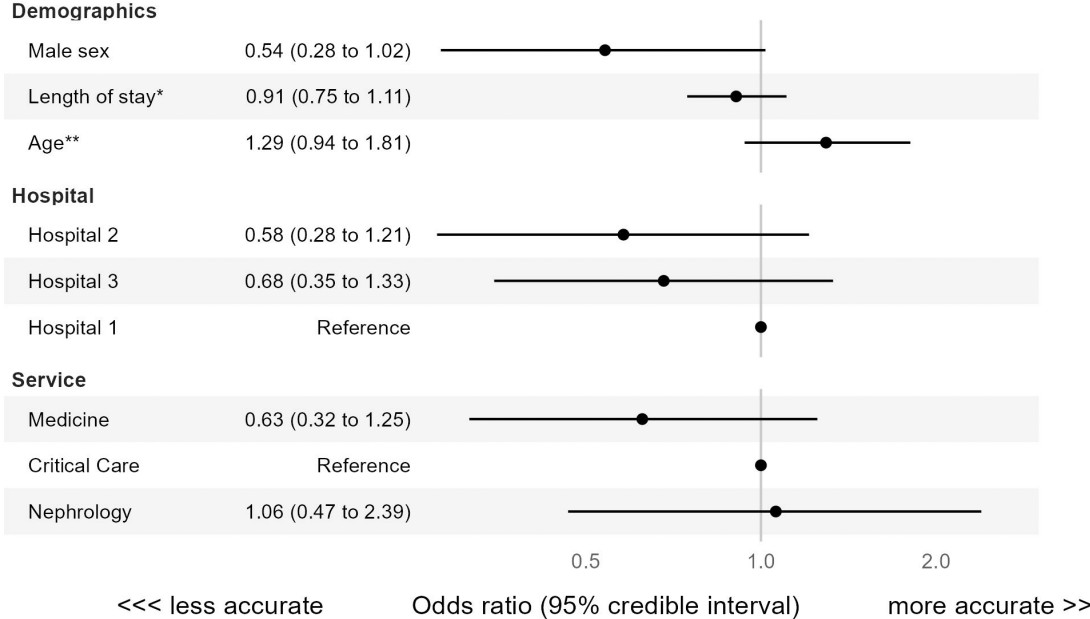

**Fig 1. This figure shows a forest plot of odds ratios for correctly identifying EHR-based non-English language preference.** This model was fit using all patients with an interview-based non-English language preference, and also included random intercepts by non-English preferred language (Fig 2). The point estimate for each odds ratio is shown as a black dot, with a horizontal black line denoting the width of the 95% credible interval. Odds ratios above 1 denote factors associated with better EHR language preference accuracy, while odds ratios below 1 denote factors associated with worse EHR language preference accuracy. * We modeled the log (base 2) of length of stay, which means that the odds ratio corresponds to each doubling of the length of stay. **Age in decades was modeled as a linear predictor.

## Discussion

In this prospective point-prevalence audit, we interviewed 323 patients across 3 hospitals and found that the EHR correctly captured patient preferred language in 86% of all patients, and 69% of patients who preferred a non-English language. Accuracy for patients who preferred non-English languages was worse for patients who were younger or male, varied by hospital site and most responsible service, and did not vary by non-English preferred language. These results have implications for research investigating differences according to preferred language, highlight an opportunity to improve preferred language data, and underscore the challenges of providing language-appropriate care in linguistically diverse patient populations.

Prior work assessing the accuracy of EHR preferred language data had similar findings to our study. An audit in a large ambulatory care organization in California found an accuracy of 94% [20]. A secondary analysis of a tobacco cessation trial in Massachusetts showed that 79% of those who preferred Spanish had that preference correctly noted in the EHR [9]. A prior audit of two academic hospitals in Toronto, Canada showed sensitivities of 81% and 12% respectively [8]. Similar to past studies, our results highlight the importance of quality assurance in routinely collected data that address the social determinants of health. This is relevant for clinical care, but also relevant for operational improvements, decision support analytics, artificial intelligence, and other analyses that incorporate this data.

Our results suggest that research showing differences according to preferred language may underestimate the effect sizes of their findings [21–23]. Misclassifying patients who prefer a non-English language, which occurred in 31% of such patients in our study, would bias the findings of such studies towards the null. This means that current research may

## Odds of correctly identifying non-English preference by language

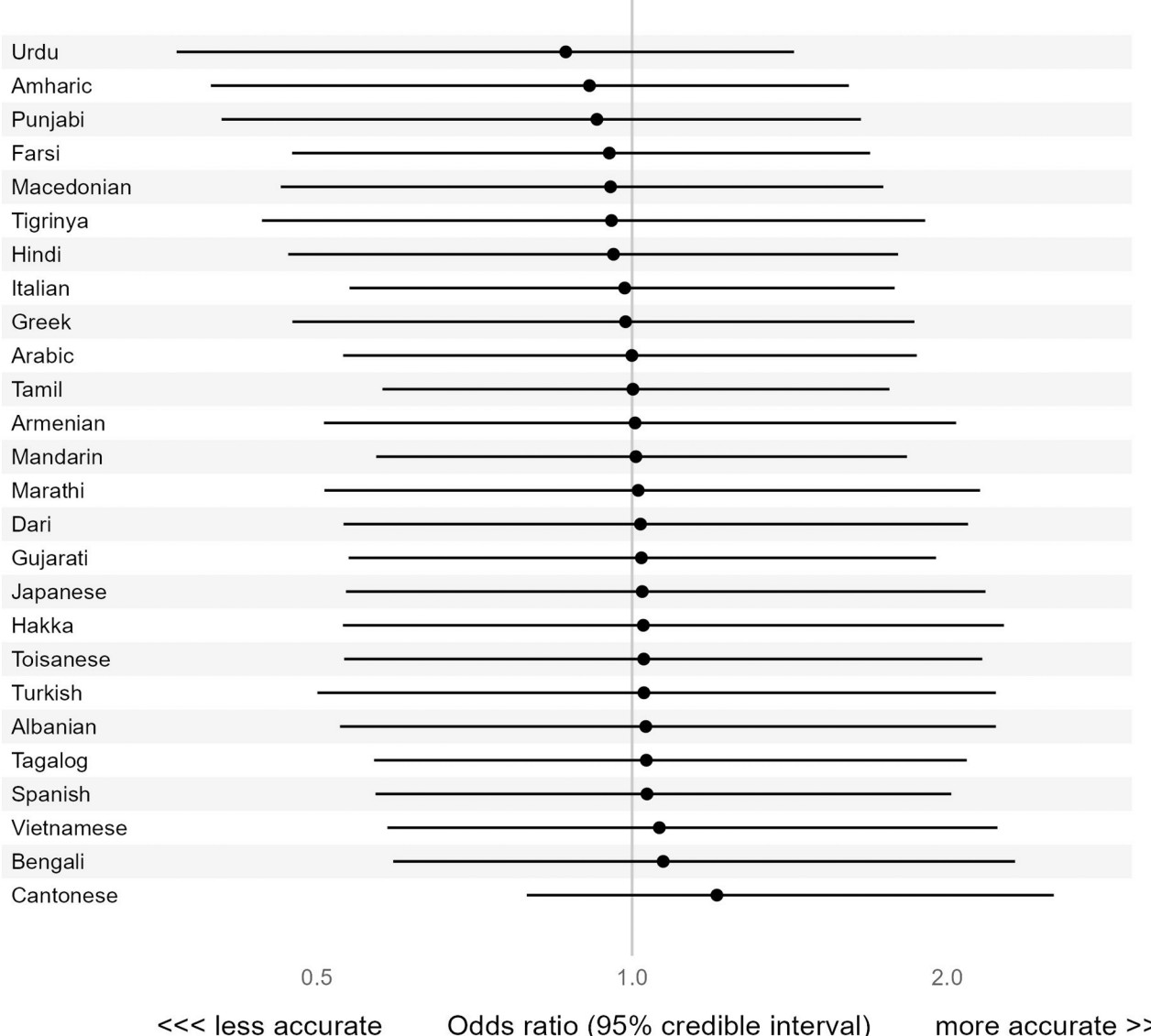

**Fig 2. Association between non-English preferred language and the probability of the EHR correctly identifying non-English preference.**
Caption: This figure shows a forest plot of odds ratios for EHR-based non-English language preference, showing the value of the random intercepts. This model was fit on all patients with an interview-based non-English language preference, and also included fixed predictors (Fig 1). The point estimate for each odds ratio is shown as a black dot, with a horizontal black line denoting the width of the 95% credible interval. Odds ratios above 1 denote factors associated with better EHR language preference accuracy, while odds ratios below 1 denote factors associated with worse EHR language preference accuracy. Note that no languages are convincingly associated with higher or lower probability of correctly identifying non-English preference.

underestimate the impact of differences in care for people with non-English preferred language. Misclassification could also impede efforts to correct algorithms and avoid algorithmic bias [24].

Our findings suggest opportunities to improve documentation of preferred language in the EHR. The reasons for lower accuracy among male compared to female patients, and younger compared to older patients, are not available in our

data, but could relate to the involvement of additional family members. Further inquiry in this area, and efforts to assess if these findings replicate at other sites, would be valuable. We also found lower accuracy in patients admitted to internal medicine and variation by hospital site. Patients admitted to nephrology, compared to internal medicine, may have had more prior interaction with the health network, allowing for more opportunities to document preferred language. Differences among hospitals could relate to differences in patient registration practices. Strategies to address these differences include ensuring front-line administrative staff, patients, and families are able to update preferred language data, and ensuring visibility of the data across the health system [20]. Front-line administrative staff can ask about language preference in a standardized manner, using a question similar to the one used in this audit [25].

Improving accurate documentation of non-English language preference is important to enable language-appropriate care. Accurate measures will also enable health systems to incorporate sociodemographic variables into care delivery improvement interventions. When such measures are not considered, health systems risk widening equity-related outcome disparities [26]. For example, when patient portals were introduced to improve asthma care among low-income patients in New York, uptake was disproportionately low among high-risk elderly and non-English speaking patients [27]. Implementing new strategies to improve preferred language data collection could help ensure each patient receives language-appropriate care and provide a tool for incorporating equity measures into health system quality improvement activities [28].

Among the 124 patients who preferred a non-English language, there were 26 different preferred languages. This resulted in small group sizes for each individual non-preferred language, which limited our certainty regarding language-specific variation in accuracy. In other environments there may be a single majority language among those who prefer a non-English language, such as Spanish in some US centers. Some studies evaluating outcomes by language preference specifically compare patients who prefer English to patients who prefer Spanish [29], while others study interventions designed solely for Spanish-speaking patients [9]. The same strategies or designs may not perform similarly in health care environments where there is a greater diversity of non-English preferred languages.

The need for interpretation in many different languages complicates providing language-appropriate care, because it limits the ability of an organization to hire full-time interpreters, complicates the development of translated non-English resources, and reduces the return-on-investment for clinicians who make efforts to learn an additional language. By contrast, telephone, video, and app-based professional interpretation services may be more helpful, due to the access to many different languages at any given time [30,31]. In the absence of professional interpreters, "ad hoc" interpreters such as family members, friends, or bilingual hospital staff often step into the medical interpretation role [32,33]. While such interpreters have the benefit of being easily accessible, professionally trained interpreters commit fewer communication errors and preserve confidentiality [34,35]. Collecting accurate preferred language data on an ongoing basis can help organizations decide which combinations of interpretation services will best suit their needs. Data on fluency of healthcare providers is also important [4].

## Limitations

Dichotomizing patients' language preference into English and non-English groups did not account for the dynamic nature of language preference or for patients who might be multilingual. We did not account for individuals who may be comfortable discussing some aspects of their care in English (such as symptoms or historical details) but not others (such as advanced care planning discussions). Language preference in an English-speaking healthcare setting is driven by a complex mix of factors in addition to English proficiency, including cultural identity, perceived appropriateness of requesting interpretation, and the logistical availability of language services. This means that language preference is a limited proxy for communication vulnerability. In addition, language preference is a subjective construct, so the absence of complementary data such as interpreter use, health literacy, or communication satisfaction with respect to the health care team limit the interpretability of the results.

We did not account for secondary language loss, which occurs when patients with transient or permanent neurological disease or injury revert to their first language [36]. We treated language preference as fixed across all healthcare encounters, which may not reflect reality for some patients who are comfortable in English for some encounters but would prefer another language for other encounters. For patients who were unable to provide a response, we relied on substitute decision makers to indicate preference, whose response may differ from the patient's true response. Our sample size limits conclusions regarding accuracy according to non-English preferred language. Generalizability of this audit may be limited by the single-month duration, sampling of specific hospital services, potential site-level variation, and the unique linguistic diversity of Toronto's population.

## Conclusions

In this point-prevalence audit of a multi-hospital system, we found that the EHR accurately captured non-English language preference for 86% of all patients and 69% of patients who preferred a non-English language. This degree of misclassification could lead to underestimation of the association between non-English language preference and processes or outcomes of care, and could impede efforts to avoid algorithmic bias. Next steps include analyzing the process of data collection and entry to improve data quality, and investigation of differences in care processes or outcomes according to preferred language.

## Author contributions

**Conceptualization:** Camron D. Ford, Thomas Bodley, Rob A. Fowler, Michele James, Paul Tam, George Tomlinson, Christopher J. Yarnell.

**Data curation:** Camron D. Ford, Christopher J. Yarnell.

**Formal analysis:** Camron D. Ford, George Tomlinson, Christopher J. Yarnell.

**Funding acquisition:** Martin Betts, Rob A. Fowler, Michele James, Paul Tam, George Tomlinson, Christopher J. Yarnell.

**Investigation:** Camron D. Ford, George Tomlinson, Christopher J. Yarnell.

**Methodology:** Camron D. Ford, Thomas Bodley, Rob A. Fowler, Shail Rawal, Christina Reppas-Rindlisbacher, George Tomlinson, Christopher J. Yarnell.

**Project administration:** Thomas Bodley, Rob A. Fowler, Alexis Gordon, Michele James, Christopher J. Yarnell.

**Resources:** Thomas Bodley, Martin Betts, Rob A. Fowler, Alexis Gordon, Michele James, Shail Rawal, Paul Tam, Christopher J. Yarnell.

**Software:** Christopher J. Yarnell.

**Supervision:** Thomas Bodley, Martin Betts, Rob A. Fowler, Alexis Gordon, Shail Rawal, Christina Reppas-Rindlisbacher, Paul Tam, George Tomlinson, Christopher J. Yarnell.

**Visualization:** Christopher J. Yarnell.

**Writing – original draft:** Camron D. Ford, Christopher J. Yarnell.

**Writing – review & editing:** Camron D. Ford, Thomas Bodley, Martin Betts, Rob A. Fowler, Alexis Gordon, Michele James, Shail Rawal, Christina Reppas-Rindlisbacher, Paul Tam, George Tomlinson, Christopher J. Yarnell.

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
