## [Decision Letter · Decision Letter 0]

18 May 2025

PDIG-D-25-00145Accuracy of preferred language data in a multi-hospital electronic health record in Toronto, CanadaPLOS Digital Health Dear Dr. Yarnell, Thank you for submitting your manuscript to PLOS Digital Health. After careful consideration, we feel that it has merit but does not fully meet PLOS Digital Health's publication criteria as it currently stands. Therefore, we invite you to submit a revised version of the manuscript that addresses the points raised during the review process.Please submit your revised manuscript within 30 days Jun 17 2025 11:59PM. If you will need more time than this to complete your revisions, please reply to this message or contact the journal office at digitalhealth@plos.org. Please include the following items when submitting your revised manuscript:* A rebuttal letter that responds to each point raised by the editor and reviewer(s). You should upload this letter as a separate file labeled 'Response to Reviewers '. This file does not need to include responses to any formatting updates and technical items listed in the 'Journal Requirements' section below.* A marked-up copy of your manuscript that highlights changes made to the original version. You should upload this as a separate file labeled 'Revised Manuscript with Track Changes '.* An unmarked version of your revised paper without tracked changes. You should upload this as a separate file labeled 'Manuscript '.If you would like to make changes to your financial disclosure, competing interests statement, or data availability statement, please make these updates within the submission form at the time of resubmission. Guidelines for resubmitting your figure files are available below the reviewer comments at the end of this letter.We look forward to receiving your revised manuscript.Kind regards, Erika OngAcademic EditorPLOS Digital Health Erika OngAcademic EditorPLOS Digital Health Leo Anthony CeliEditor-in-ChiefPLOS Digital Healthorcid.org/0000-0001-6712-6626**Journal Requirements:**1. Your current Financial Disclosure states, “Canadian Critical Care Trials Group Network of Networks Summer Student Award (CF), Sunnybrook Program to Access Research Knowledge (CF), the Nephrology Program at the Scarborough Health Network (CF). JP Bickell Foundation Medical Research Grant (CJY). Mak Pak Chiu and Mak-Soo Lai Hing Chair in General Internal Medicine, University of Toronto (SR)”. However, your funding information on the submission form indicates that you received funding from “Canadian Critical Care Trials Group Network of Networks, J.P. Bickell Foundation and Scarborough Health Network Department of Nephrology”. Please indicate by return email the full and correct funding information for your study and confirm the order in which funding contributions should appear. Please be sure to indicate whether the funders played any role in the study design, data collection and analysis, decision to publish, or preparation of the manuscript. 2. We have amended your Competing Interest statement to comply with journal style. We kindly ask that you double check the statement and let us know if anything is incorrect. 3. Please upload a copy of Figure 1 and 2 which you refer to in your text on page 17 and 18. Or, if the figure is no longer to be included as part of the submission please remove all reference to it within the text. 4. Please provide separate figure files in .tif or .eps format. For more information about figure files please see our guidelines:  https://journals.plos.org/digitalhealth/s/figures https://journals.plos.org/digitalhealth/s/figures#loc-file-requirements**Additional Editor Comments (if provided):****Reviewers' Comments:**Reviewer's Responses to Questions

**Comments to the Author**

1. Does this manuscript meet PLOS Digital Health’s publication criteria ? Is the manuscript technically sound, and do the data support the conclusions? The manuscript must describe methodologically and ethically rigorous research with conclusions that are appropriately drawn based on the data presented.

Reviewer #1: Yes

Reviewer #2: Yes

Reviewer #3: Yes

2. Has the statistical analysis been performed appropriately and rigorously?

Reviewer #1: Yes

Reviewer #2: Yes

Reviewer #3: Yes

3. Have the authors made all data underlying the findings in their manuscript fully available (please refer to the Data Availability Statement at the start of the manuscript PDF file)?

Reviewer #1: Yes

Reviewer #2: Yes

Reviewer #3: Yes

4. Is the manuscript presented in an intelligible fashion and written in standard English?

Reviewer #1: Yes

Reviewer #2: Yes

Reviewer #3: Yes

5. Review Comments to the Author

Reviewer #1: The manuscript investigates the validity of EHR language preference data. The study design and analytical approach are appropriate, and the results shed light on areas for improvement in data capture.

I suggest the authors consider the following comments to strengthen the manuscript:

• The authors should provide more details on the prospective audit’s sampling strategy. It would help to clarify whether patients were selected consecutively or randomly during the audit period and whether any inclusion/exclusion criteria were applied. The manuscript notes that most audited patients were from internal medicine wards – if the audit was limited to certain services or a short time frame, discuss how that might affect generalizability. Ensuring the reader understands how the 323 patients were chosen (and if any patients approached declined participation or were unavailable) will bolster confidence that the sample is representative of the hospital populations.

• The handling of patient non-response needs more explanation. Nearly 41% of the interviews were conducted with substitute decision-makers, which is a pragmatic solution for incapacitated patients but could introduce uncertainty in the “true” preferred language. The manuscript should clarify if any eligible patients could not be interviewed at all and how those were handled (e.g., excluded or replaced). It would also be worthwhile to comment on whether the agreement between EHR and the interview differed when the information came from a substitute decision-maker versus the patient directly. If data permits, consider a brief analysis or comment in the Discussion on whether proxy responses might affect the accuracy.

• The use of sensitivity, specificity, PPV, and NPV to quantify the accuracy of EHR language data is appropriate for the dichotomized outcome (non-English vs. English preference). It may be useful to report an overall agreement statistic beyond simple percentage agreement; for instance, a Cohen’s kappa could contextualize the 86% agreement by accounting for the large proportion of English-preference patients. Additionally, please clarify the statistical test used for comparing the prevalence of non-English preference between interviews and EHR (the 9.6% absolute difference). A McNemar’s test would be appropriate for this paired comparison – if that was used, stating it explicitly would be helpful for readers.

• The secondary analysis using a Bayesian multilevel logistic regression model is a strength, allowing the authors to account for clustering by language and to estimate effects despite some languages having few patients. The manuscript should expand on the modeling decisions to improve transparency. The authors could specify the prior distributions used for parameters in the Bayesian model (even if default weakly-informative priors were used) and note whether any prior sensitivity analysis was performed. Additionally, consider commenting on model diagnostics: for example, whether the Markov Chain Monte Carlo converged well (R-hat values ~1, adequate effective sample sizes) – even a brief mention would reinforce confidence in the Bayesian results. Finally, given the multi-hospital setting, the decision to model hospital and service as fixed effects (with odds ratios for each) rather than random effects is acceptable, but it may help to briefly justify this choice (e.g., treating the three specific hospitals as fixed factors of interest due to known policy differences).

• In reporting the results, the authors appropriately highlight the top five non-English languages and group the rest as “other”. It might be useful to provide some additional insight (perhaps in a Supplement) into language-specific accuracy if possible – for instance, were there any notable trends such as certain languages consistently being missed by the EHR?

• The current results indicate wide credible intervals for individual language effects, suggesting no statistically significant outliers. However, the authors need to ensure that the text does not over-interpret any single language’s result given the small sample sizes. The authors may also consider explicitly stating that very low-frequency languages were included in the model via random effects, which helps borrow strength from the larger groups – this could be highlighted as a benefit of the Bayesian multilevel approach.

Reviewer #2: This is a well-executed, timely, and important study addressing the accuracy of preferred language data in a multi-hospital EHR system. It has strong methodological rigour and relevance, particularly in the context of health equity and mitigating algorithmic bias. The results are clearly presented and well-supported by the data.

Strengths:

1. Comprehensive audit across three hospitals with a robust sample size.

2. Use of Bayesian logistic regression is appropriate, particularly to handle small language clusters.

3. Strong discussion contextualising findings within prior literature and implications for research, health system improvement, and patient care.

4. Transparent data availability and ethical considerations.

Suggestions for Improvement:

1. It may be helpful to briefly explain why ICU, nephrology, and internal medicine services were specifically chosen for inclusion in the study. This additional context would allow readers to better assess the generalisability of the findings to other hospital services or patient populations.

2. Although the introduction is well written, it may benefit from highlighting Toronto’s well-known linguistic diversity. This would further underscore the relevance and uniqueness of the setting and strengthen the rationale for studying preferred language accuracy in this particular health system.

3. In Table 2, changing “Preferred Spoken Language” to “Interview-Based Preferred Language” may enhance clarity.

4. Consider adding more detail on how frontline staff collect and enter language preference data during registration, as this could give further insight into variation across hospitals.

5. Consider noting the short study period (approximately one month) as a potential limitation and whether seasonal admission patterns could influence generalisability.

Overall, this is a valuable contribution to the field of digital health and health equity, and I recommend acceptance pending minor revision.

Reviewer #3: Thank you for the opportunity to review this interesting study. The researchers conducted an audit of EHR preferred language by interviewing patients and asking the what is their preferred language for health care communication.

Overall, this is a valuable contribution. I hope the authors find my comments and suggestions helpful to improve the article.

Introduction

- The sentence that states “Language concordance, via shared language proficiency or interpretation” should be rephrased for accuracy. Interpreter-mediated care is not language-concordant. There is still a discordance between the clinician and patient during interpreter-mediated care. However, it is language-appropriate because the interpreter serves as the linguistic conduit to negotiate meaning between the patient and clinician. The authors could consider saying “Language-appropriate care, via shared language proficiency or interpretation” or could consider other adjustments for clarity.

- The authors do a good job of framing the importance of language preference data collection.

- The authors explain prior related studies on the accuracy of preferred language data and explain the gap that this study aims to address. It seems to me that there are other things that distinguish this study from the two that were cited, references 8 and 9. For example, ref 8 is a US study and therefore reflects a linguistically different population compared to Canada. It may be worth pointing this out to add to this study’s contribution to the literature.

Methods

- The authors describe the researcher going into each room and asking the scripted question. However, if the patient had a non-English language preference, how was the question asked? Was there a protocol for when an interpreter was used to ask the question (e.g., based on what was documented in the EHR? Something else?). What types of interpreters were used to ask the question? If interpreters were not used, then what other approaches were used to ensure the question and response were communicated accurately?

- The methods state that if “a response could not be elicited”, a substitute decision-maker was contacted. What does it mean for the response to not be able to be elicited? Was it based on the patient’s mental status, language/inability to communicate, refusal, or a combination of reasons? Some explanation here would be important.

Results

- The authors should move the first several sentence of the discussion into the results.

Discussion

- The discussions should begin with the major findings summary but without repeating the findings themselves that were already presented in the results.

- The authors do a good job explaining the risks of misclassification of patients who have non-English language preference.

- Is there also room to discuss the dynamic nature of language preference? Could this be part of the reason that there is a discrepancy, and that some amount of discrepancy in this data point may be unavoidable? My meaning is that depending on the context, a patient may have differing preferences with regards to the language they would want to use in a particular health care encounter. For example, if a person has basic English skills but generally prefers their health care in Cantonese, is it possible that this person would choose Cantonese if they had a Cantonese speaking doctor but English if they had an English-speaking doctor who would need to use an interpreter if they were to hold the encounter in Cantonese? Some more nuanced discussion of the dynamic nature of language preference and impact of multilingualism in the community would be warranted and useful.

- Perhaps related to my latter point, what is the authors’ interpretation of the finding that older patients’ language preference was more reliably documented in the EHR compared to younger patients? I can think of several possibilities, and it may be useful for the authors to share some potential thoughts and perhaps a future research agenda that could test the potential factors involved.

- Why do the authors think the accuracy rate of the language data field varied across languages? Could there be cultural, historical, or political factors that affect how the data is reported/collected or how patients respond?

- The authors could include a call for the collection of language proficiency data about health care professionals, which is also needed to conduct language concordance research. See, for example, Ortega et al. JAMA Network Open: https://jamanetwork.com/journals/jamanetworkopen/fullarticle/2807144.

- The authors provide some practical recommendations in the discussion, such as the tips for registration staff. I would welcome a table or chart that summarizes the key strategies/recommendations the authors have for health systems to implement a better system for collecting language preference. Are there quality improvement/audit metrics that should be used, and if so, with what frequency, to ensure the accuracy of the EHR language preference? How should the language preference question be asked by clerks registering patients?

Limitations

- It is unclear what the authors mean by this sentence: “We did not account for bilingual or moderately fluent individuals.”

- Additional limitations should be considered such as how generalizable (or not) these results might be in other types of hospitals/health systems, cities, countries, etc.

- Consider adding suggestions for future study that could address remaining questions.

6. PLOS authors have the option to publish the peer review history of their article (what does this mean? ). If published, this will include your full peer review and any attached files.

**Do you want your identity to be public for this peer review?** For information about this choice, including consent withdrawal, please see our Privacy Policy .

Reviewer #1: No

Reviewer #2: No

Reviewer #3: **Yes: ** Pilar Ortega

**Figure resubmission:** While revising your submission, please upload your figure files to the Preflight Analysis and Conversion Engine (PACE) digital diagnostic tool, https://pacev2.apexcovantage.com/. PACE helps ensure that figures meet PLOS requirements. To use PACE, you must first register as a user. Registration is free. Then, login and navigate to the UPLOAD tab, where you will find detailed instructions on how to use the tool. If you encounter any issues or have any questions when using PACE, please email PLOS at figures@plos.org. Please note that Supporting Information files do not need this step. If there are other versions of figure files still present in your submission file inventory at resubmission, please replace them with the PACE-processed versions.**Reproducibility:** To enhance the reproducibility of your results, we recommend that authors of applicable studies deposit laboratory protocols in protocols.io, where a protocol can be assigned its own identifier (DOI) such that it can be cited independently in the future. Additionally, PLOS ONE offers an option to publish peer-reviewed clinical study protocols. Read more information on sharing protocols at https://plos.org/protocols?utm_medium=editorial-email&utm_source=authorletters&utm_campaign=protocols

---

## [Decision Letter · Decision Letter 1]

29 Jul 2025

PDIG-D-25-00145R1Accuracy of preferred language data in a multi-hospital electronic health record in Toronto, CanadaPLOS Digital Health Dear Dr. Yarnell, Thank you for submitting your manuscript to PLOS Digital Health. After careful consideration, we feel that it has merit but does not fully meet PLOS Digital Health's publication criteria as it currently stands. Therefore, we invite you to submit a revised version of the manuscript that addresses the points raised during the review process. Please submit your revised manuscript within 30 days Aug 28 2025 11:59PM. If you will need more time than this to complete your revisions, please reply to this message or contact the journal office at digitalhealth@plos.org. Please include the following items when submitting your revised manuscript:* A rebuttal letter that responds to each point raised by the editor and reviewer(s). You should upload this letter as a separate file labeled 'Response to Reviewers '. This file does not need to include responses to any formatting updates and technical items listed in the 'Journal Requirements' section below.* A marked-up copy of your manuscript that highlights changes made to the original version. You should upload this as a separate file labeled 'Revised Manuscript with Track Changes '.* An unmarked version of your revised paper without tracked changes. You should upload this as a separate file labeled 'Manuscript '. If you would like to make changes to your financial disclosure, competing interests statement, or data availability statement, please make these updates within the submission form at the time of resubmission. Guidelines for resubmitting your figure files are available below the reviewer comments at the end of this letter. We look forward to receiving your revised manuscript. Kind regards, Gloria Hyunjung KwakSection EditorPLOS Digital Health Gloria Hyunjung KwakSection EditorPLOS Digital Health Leo Anthony CeliEditor-in-ChiefPLOS Digital Healthorcid.org/0000-0001-6712-6626  **Journal Requirements:** If the reviewer comments include a recommendation to cite specific previously published works, please review and evaluate these publications to determine whether they are relevant and should be cited. There is no requirement to cite these works unless the editor has indicated otherwise.  **Additional Editor Comments (if provided):** Thank you for your thoughtful revision. I have one additional suggestion regarding the interpretation of the “preferred language ≠ English” variable.

In English-speaking healthcare settings, this variable can represent a complex mix of factors beyond limited English proficiency—including bilingualism, cultural identity, administrative choices during intake, and the influence of available language services. Patients may select a non-English language due to family context or offered support, not necessarily because of communication difficulty.

Additionally, “language comfort” is a subjective construct that is difficult to assess uniformly, and the absence of complementary indicators such as interpreter use, health literacy, or communication satisfaction may limit interpretability.

For these reasons, I encourage you to acknowledge these limitations in the Discussion or Limitations section to help contextualize the role of language preference and avoid overgeneralizing this as a proxy for communication vulnerability.

Thank you again for your contribution.**Reviewers' Comments:** Reviewer's Responses to Questions

**Comments to the Author**

1. If the authors have adequately addressed your comments raised in a previous round of review and you feel that this manuscript is now acceptable for publication, you may indicate that here to bypass the “Comments to the Author” section, enter your conflict of interest statement in the “Confidential to Editor” section, and submit your "Accept" recommendation.

Reviewer #1: All comments have been addressed

Reviewer #2: (No Response)

Reviewer #3: All comments have been addressed

2. Does this manuscript meet PLOS Digital Health’s publication criteria ? Is the manuscript technically sound, and do the data support the conclusions? The manuscript must describe methodologically and ethically rigorous research with conclusions that are appropriately drawn based on the data presented.

Reviewer #1: Yes

Reviewer #2: Yes

Reviewer #3: Yes

3. Has the statistical analysis been performed appropriately and rigorously?

Reviewer #1: Yes

Reviewer #2: Yes

Reviewer #3: Yes

4. Have the authors made all data underlying the findings in their manuscript fully available (please refer to the Data Availability Statement at the start of the manuscript PDF file)?

Reviewer #1: Yes

Reviewer #2: Yes

Reviewer #3: Yes

5. Is the manuscript presented in an intelligible fashion and written in standard English?

Reviewer #1: Yes

Reviewer #2: Yes

Reviewer #3: Yes

6. Review Comments to the Author

Reviewer #1: I appreciate the time and effort that the authors have put into addressing the feedback and criticisms.

I agree withe changes made and have no further comments.

Reviewer #2: Thank you for addressing the comments I raised. Just a minor correction (Line 26-27): there appears to be a grammatical issue in the revised sentence—“have a mother-tongue other English or French.” To improve clarity and correctness, consider changing it to “have a mother tongue other than English or French.” Also, the hyphen in “mother tongue” is unnecessary.

Reviewer #3: Thank you for thoughtfully addressing the reviewer comments.

7. PLOS authors have the option to publish the peer review history of their article (what does this mean? ). If published, this will include your full peer review and any attached files.

**Do you want your identity to be public for this peer review?** For information about this choice, including consent withdrawal, please see our Privacy Policy .

Reviewer #1: No

Reviewer #2: No

Reviewer #3: **Yes: ** Pilar Ortega, MD, MGM

---

## [Editor Report · Decision Letter 2]

15 Aug 2025

Accuracy of preferred language data in a multi-hospital electronic health record in Toronto, Canada

PDIG-D-25-00145R2

Dear Dr. Yarnell,

We're pleased to inform you that your manuscript has been judged scientifically suitable for publication and will be formally accepted for publication once it meets all outstanding technical requirements.

Within one week, you'll receive an e-mail detailing the required amendments. When these have been addressed, you'll receive a formal acceptance letter and your manuscript will be scheduled for publication.

An invoice for payment will follow shortly after the formal acceptance. To ensure an efficient process, please log into Editorial Manager at https://www.editorialmanager.com/pdig/ click the 'Update My Information' link at the top of the page, and double check that your user information is up-to-date. For billing related questions, please contact billing support at https://plos.my.site.com/s/.

Kind regards,

Sarah Mayo

Staff Admin

PLOS Digital Health